# Spectrum of Thalassemia and Hemoglobinopathy Using Capillary Zone Electrophoresis: A Facility-Based Single Centred Study at icddr,b in Bangladesh

Anamul Hasan [1] , Jigishu Ahmed [2] , Bikash Chandra Chanda [1], Maisha Aniqua [3], Raisa Akther [1], Palash Kanti Dhar [1], Kazi Afrin Binta Hasan [1], Abdur Rouf Siddique [1], Md. Zahidul Islam [1], Sharmine Zaman Urmee [1] and Dinesh Mondal [1],*

1   Clinical Hematology and Cancer Biology Laboratory, Laboratory Sciences and Services Division, International Centre for Diarrhoeal Disease Research, Bangladesh (icddr,b), Dhaka 1212, Bangladesh; nmlhsn@gmail.com (A.H.)
2   Laboratory of Gut-Brain Signaling, Laboratory Sciences and Services Division, International Centre for Diarrhoeal Disease Research, Bangladesh (icddr,b), Dhaka 1212, Bangladesh
3   Mechanobiology Laboratory, Department of Biochemistry and Molecular Biology, University of Texas Medical Branch, Galveston, TX 77555, USA
*   Correspondence: din63d@icddrb.org

**Abstract:** Background: Although the global thalassemia zone covers Bangladesh, there are very limited studies conducted in this region. Therefore, the focus of our study is to understand the prevalence and burden of thalassemia and hemoglobinopathy in Bangladesh. Methods: The analysis was based on a retrospective evaluation of laboratory diagnoses between 2007 January and 2021 October. A total of 8503 specimens were sampled and analyzed which were either referred by corresponding physicians or self-referred. This was neither any epidemiological nationwide survey nor was the study population chosen randomly. Hematological data were obtained through capillary zone electrophoresis and corresponding complete blood count. Results: 1971 samples (~23.18% of the total) were found with at least one inherited hemoglobin disorder. The most common hemoglobin disorder observed was the hemoglobin E (Hb E) trait (10.67%), followed by the β-thalassemia trait (8.4%), homozygotic Hb E (1.59%), and Hb E/β-thalassemia (1.58%). Other variants found in this study with minimal percentages were Hb N-Seattle, Hb S, Hb D-Punjab, Hb Lepore, Hb C, Hb Hope, Hb H, and hereditary persistence of fetal hemoglobin. Discussion: The pattern of thalassemia and hemoglobinopathy in our study is diverse and heterogeneous. A broad and detailed spectrum of such inherited hemoglobin disorders will ultimately be helpful in implementing nationwide thalassemia management and strategy policy in Bangladesh.

**Keywords:** thalassemia; hemoglobinopathy; Bangladesh; prevalence; hemoglobin; electrophoresis; hemoglobin variants; epidemiology





## 1. Introduction

Inherited hemoglobin disorders are the most common non-communicable autosomal recessive blood disorder posing major public health concerns in many parts of the world including Bangladesh [1–4]. Inherited hemoglobin disorders can be qualitative (or structural) irregularities having different types of mutations [5] or can be quantitative (or regulation) abnormalities resulting in a decreased or unbalanced synthesis of at least one globin polypeptide chain (alpha, α; beta, β; gamma, γ or delta, δ) in hemoglobin (Hb) tetramer [6,7]. Structural irregularities are responsible for hemoglobinopathies whereas thalassemias result from quantitative defects in globin chain production [8]. Hemoglobinopathies and thalassemias with high prevalence in tropical and subtropical areas covering Africa, Southeast Asia and the Mediterranean basin are now globally distributed due to increased migratory patterns and lack of preventive awareness among suspected populations [9,10].

As the global thalassemia zone covering tropical and sub-tropical areas includes Bangladesh, the prevalence and epidemiological burden of hemoglobinopathies and thalassemias in this country are essential to consider [1,9]. In Bangladesh, 1–5% (average 3.35%) population are β-thalassemia carriers while about 10% are carrying the Hb E trait [11]. However, the prevalence of the Hb E trait is higher (~36%) in endogenous tribal groups in Bangladesh [12]. According to Thalassemia International Foundation, the birth incidence of having thalassemia and hemoglobinopathy syndromes is 350 in 1 million population which is particularly higher in tribal populations (600 in 1 million) in Bangladesh. However, due to the lack of population-based data and a national patient registry, the exact detailed information on the epidemiological burden of thalassemia and hemoglobinopathy in Bangladesh is inadequate [11].

In the coming days with the rapidly increasing population, thalassemias and other hemoglobinopathies will be a major public health issue in Bangladesh [13]. Most of the children are neither diagnosed nor treated resulting in death in early infancy due to anemia. Tertiary-level treatment is only available in the capital city, Dhaka and in the port city, Chittagong. Moreover, the costs of treatment are beyond the capabilities of the majority of families. Only 1% of the thalassemia population receives benefits from reasonable management [11]. There is no established curative therapy for thalassemia except bone marrow transplantation which is very expensive with significant rejection risk [14]. Hence, a 14-year-long spectrum of inherited blood disorders has been conducted at International Centre for Diarrhoeal Disease Research, Bangladesh (icddr,b) as a facility-based study which will be helpful for the prevention of thalassemia at the national level.

## 2. Materials and Methods

### 2.1. Study Design

The analysis is based on a retrospective evaluation of laboratory diagnoses made from January 2007 to September 2021 (inclusive) with a timeframe of more than 14 years. During that timeframe, 8503 samples were collected and analyzed. Blood samples preserved with EDTA were either referred by corresponding physicians or self-referred. Secondary data analysis was performed using primary data collected from the activity program (Protocol no: PR-22053 version: 01), housed within the laboratory of Clinical Hematology and Cancer Biology, icddr,b. All samples requiring thalassemia and hemoglobinopathy screenings were included in the study. However, those with a recent history of transfusion (three months prior to sample collection) were excluded from the study. The specimens identified with any discordant hemogram and electrophoresis patterns were contacted (via telephone) in order to ascertain about the recent history of blood transfusion. Assigned laboratory physicians at Clinical Hematology and Cancer Biology laboratory, icddr,b did the follow-up as a part of necessary laboratory practices (for those particular cases) to get the best clinical correlation. Sometimes, in very few cases like Hb S/β-thalassemia, Hb E/Hb S, Hb D-Punjab/β-thalassemia, Hb D-Punjab/Hb S, Hb N-Seattle, etc., we also analyzed the samples from their first-degree relatives (individual's parents or siblings or offsprings) to confirm our findings of rare Hb variants. Besides, genotype study (reverse dot-blot hybridization) and high-performance liquid chromatography (HPLC) were done whenever required to clinically correlate our outcomes obtained from capillary electrophoresis analysis.

### 2.2. Ethical Consideration

The dataset was designed in a de-identified format. The identities (name, sex, age, ethnicity, religion, whereabouts) of all samples were kept anonymous. A protocol was strictly followed to protect the privacy and rights of every sample. The study team fulfilled administrative and ethical approval requirements. The research protocol (PR-22053) of this study was approved as review exemption status by the Institutional Review Board (IRB) othe f Research Administration of icddr,b.

### 2.3. Sample Preparation and Analysis

A total of 3 mL of blood samples were collected in BD Vacutainer® (anticoagulant K2EDTA 5.4 mg, Ref: 367856) at the specimen reception unit, icddr,b. However, few samples were collected outside the designated specimen reception units of icddr,b. Collected samples if not analyzed immediately were stored for up to 4 days between 2 to 8 °C to minimize progressive hemoglobin degradation. However, by no means was any sample stored at room temperature.

The initial screening was done by complete blood count (CBC) using two automated blood counters- Sysmex XT-1800i (Sysmex Corporation, Kobe, Japan) and Sysmex XN-1000 (Sysmex Corporation, Kobe, Japan), according to the manufacturer's guidelines [15]. Different erythrocyte parameters such as hemoglobin concentration, hematocrit (HCT), red blood cell indices like mean corpuscular volume (MCV), mean corpuscular hemoglobin (MCH), mean corpuscular hemoglobin concentration (MCHC), red cell distribution width (RDW) were obtained from the CBC results.

The final screening was performed by capillary zone electrophoresis (CZE) using a capillary electrophoresis instrument (Capillarys2, Sebia, Lisses, France) where different hemoglobin variants were quantified. During normal operations, 200 μL of packed erythrocytes of each specimen were sampled, diluted and lysed. An electropherogram of each sample was determined by normalizing Hb A and Hb A2 values of the normal Hb A2 control samples, according to the manufacturer's guidelines. Each electropherogram had color-coded distinct peaks in 15 different zones and was calculated as a percentage. Data were recorded and analyzed using Phoresis CORE (Version 2.0) [16,17]. Potential Hb variants located in 15 different zones (Z1 to Z15) of Sebia-based capillary zone electrophoresis (See Supplementary Table S1).

### 2.4. Sickling Test to Exclude Hb D-Punjab

By capillary zone electrophoresis, Hb D-Punjab can incorrectly be identified as Hb S as these two Hb variants are detected in the same area. In order to solve such a possible discrepancy, the sickling test was done with 2% sodium metabisulfite by which deoxygenation compels Hb S to become polymerized inside the red cells resulting in damage to the cell membrane [18,19].

### 2.5. Red Cell Indices and Parameter of Anemia

Mean corpuscular volume (MCV), mean corpuscular hemoglobin (MCH), mean corpuscular hemoglobin concentration (MCHC), and red cell distribution width (RDW) are the red cell indices [20]. In adults, elderly, and children, normal blood samples generally have (at least) MCV = 80 fL, MCHC = 32 g/dL, and MCH = 27 pg which are different for newborns (normal range; MCV = 96–108 fL, MCHC = 32–33 g/dL, MCH = 32–34 pg) [21]. The parameter, MCV is used to classify anemic condition based on whether the MCV is low (microcytic; <80 fL) [21], normal (normocytic; 80–100 fL) [22], or elevated (macrocytic; >100 fL) [23]. MCV and MCH with the value less than 80 fL and 27 pg, respectively, are generally suggestive of thalassemia and hemoglobinopathy [9,24]. RBC distribution width (RDW) is also used in conjunction with the MCV and MCH to narrow down the possible causes of anemia related to inherited Hb disorders. The coefficient of variation of RDW (RDW-CV) deals with the variation in size of RBCs (anisocytosis) in percentile format. RDW-CV, in normocytic conditions, varies between 12% and 17% with the consideration of age, sex, and ethnic subgroup as well [25]. A higher RDW-CV is positively correlated with a larger variation in the size of RBCs implying a more heterogenous population [26].

According to World Health Organization, anemia in children below 5 years old and pregnant females is defined when the hemoglobin concentration is <11 g/dL. Similarly the cutoff values of hemoglobins are <12 g/dL for non-pregnant females [27] and <13 g/dL for males [9]. In the case of severe anemia, cutoff values of hemoglobin concentration are <7.0 g/dL for children younger than 5 years and pregnant females and <8.0 g/dL for

non-pregnant females and males [28]. Moreover, a notable feature of severe anemia is having a hematocrit (HCT) level with less than 25% [29].

Besides thalassemia and hemoglobinopathy cases, we have determined several specimens having nutritional anemia. Globally, iron deficiency anemia is the most prevalent form of nutritional anemia seen especially in women and younger populations [30]. However, we would not discuss extensively about any type of nutritional anemia found here. Many samples with normal erythrocytic parameters and Hb variants (e.g., Hb A, Hb A2, and Hb F) were referred to as normal populations.

*2.6. Data Management*

Raw data were received and entered in Microsoft Excel format. Documented data were then analyzed and curated using STATA (v.16) (StataCorp LLC, College Station, TX, USA). Separate datasets were created and maintained for different types of cases. Every STATA dataset was thoroughly scrutinized to check for integrity, duplicate entries, inconsistency, and inaccuracy. Eventually, all STATA files were merged to create a final dataset keeping every individual dataset as backup. We have calculated the median age instead of the mean as the latter one is sensitive to extreme values named outliers [31]. Continuous data were presented as mean ± standard deviation (SD). 95% confidence interval (CI) was calculated using STATA (v.16). Graphical presentation was done using GraphPad Prism (v.9.4.0) (GraphPad Software Inc., La Jolla, CA, USA).

**3. Results**

The overview of our retrospective study is presented in Table 1. A total of 8503 specimens were sampled and analyzed. Of them, 1971 samples (23.18% of total studied) were diagnosed with at least one type of inherited hemoglobin disorder. The most common scenario observed was the hemoglobin E (Hb E) trait (46.02% from 1971 samples), followed by β-thalassemia trait, Hb E homozygote, and Hb E/β-thalassemia with 36.23%, 6.80% and 6.85% from 1971 samples, respectively. Besides β-thalassemia, Hb H (–/-α), Hb Lepore and hereditary persistence of fetal hemoglobin (HPFH), etc., were analyzed from the capillary electrophoresis.

The average percentages of Hb E were 23.8 ± 1.9 (95% CI, 23.6–23.9), 91.9 ± 2.9 (95% CI, 91.4–92.4) and 50.7 ± 12.2 (95% CI, 48.6–52.8) for Hb E (heterozygote), Hb E (homozygote) and Hb E/β-thalassemia (double heterozygote), respectively. The average percentage of Hb A2 was 5.2 ± 0.6 (95% CI, 5.1–5.3) for β-thalassemia (heterozygote). The average percentages of Hb F were 93.9 ± 3.9 (95% CI, 90.9–96.9) and 39.6 ± 14.0 (95% CI, 37.2–42.0) for β-thalassemia (homozygote) and Hb E/β-thalassemia (double heterozygote), respectively. The average percentages of other altered Hb variants like Hb Lepore, Hb D-Punjab, Hb S, Hb H, etc. are also tabulated in Table 1.

In Figure 1A–F, we performed comparative analyses among Hb E (heterozygote), Hb E (homozygote), Hb E/β-thalassemia (double heterozygote), β-thalassemia (heterozygote) and β-thalassemia (homozygote) in terms of Hb concentration, hematocrit (HCT) and red cell indices (e.g., MCV, MCH, MCHC and RDW-CV), based on the data from Table 2. Hemoglobin concentration and hematocrit level for β-thalassemia (homozygote) and Hb E/β-thalassemia were below the corresponding cutoff values of severe anemia (Figure 1A,B). Moreover, MCH and MCV values of all six types of inherited hemoglobin disorders in this study were below the corresponding cutoff values (Figure 1C,D). Mean MCHC values were in the range of 30 to 34 g/dL for all of them (Figure 1E). In terms of RDW-CV, we have found two patterns- mean value of <20% RDW-CV was seen in those samples having Hb E (Heterozygote and homozygote) and β-thalassemia heterozygote conditions, whereas mean value of >25% RDW-CV was found in other three types of conditions (Figure 1F).

**Table 1.** Overview of thalassemia and hemoglobinopathy in the study (N = 8503).

| Inherited Hb Disorders | | Frequency, *n* (%) | Hb A, % (Mean ± SD) | Hb A2, % (Mean ± SD) | Hb F, % (Mean ± SD) | Altered Hb Variants, % (Mean ± SD) | |
|---|---|---|---|---|---|---|---|
| **Heterozygote Condition** | | | | | | | |
| Hb E | | 907 (10.67) | 71.4 ± 2.3 | 3.4 ± 0.4 | 1.9 ± 2.3 | Hb E 23.8 ± 1.9 | - |
| β-thalassemia | | 714 (8.40) | 93.8 ± 1.6 | 5.2 ± 0.6 | 1.6 ± 1.6 | - | - |
| Hb D-Punjab | | 16 (0.19) | 58.7 ± 4.0 | 2.9 ± 0.4 | 1.0 ± 0.9 | Hb D-Punjab 37.2 ± 4.0 | - |
| HPFH * | | 13 (0.15) | 80.9 ± 3.3 | 2.5 ± 0.7 | 16.6 ± 3.1 | - | - |
| Hb S | | 9 (0.11) | 58.1 ± 2.0 | 2.7 ± 0.2 | 0.9 ± 0.5 | Hb S 38.6 ± 1.7 | - |
| **Double heterozygote condition** | | | | | | | |
| Hb E/β-thalassemia | Hb E/β⁺ | 101 (1.19) | 6.0 ± 2.6 | 5.1 ± 1.6 | 37.7 ± 11.8 | Hb E 50.9 ± 10.1 | - |
| | Hb E/β⁰ | 33 (0.39) | - | 4.8 ± 1.6 | 45.3 ± 18.5 | Hb E 50.0 ± 17.4 | - |
| Hb E/Hb S | | 5 (<0.1) | - | 3.0 ± 0.7 | 21.6 ± 13.3 | Hb E 23.6 ± 2.7 | Hb S 51.8 ± 11.6 |
| Hb S/β⁺-thalassemia | | 1 (<0.1) | 7.0 | 4.8 | 22.1 | Hb S 66.1 | - |
| Hb D-Punjab/β⁺-thalassemia | | 1 (<0.1) | 4.1 | 5.2 | 1.4 | Hb D-Punjab 89.3 | - |
| Hb S/Hb D-Punjab | | 1 (<0.1) | 0.9 | 2.4 | 2.6 | Hb D-Punjab 50.8 | Hb S 43.3 |
| **Homozygote condition** | | | | | | | |
| Hb E | | 135 (1.59) | 2.4 ± 2.1 | 4.6 ± 0.8 | 2.5 ± 3.0 | Hb E 91.9 ± 2.9 | - |
| β-thalassemia | | 9 (0.11) | 3.7 ± 2.7 | 2.4 ± 1.3 | 93.9 ± 3.9 | - | - |
| Hb D-Punjab | | 1 (<0.1) | - | 3.4 | 1.7 | Hb D-Punjab 94.9 | - |
| HPFH * | | 1 (<0.1) | - | - | 100 | - | - |
| **Other Hb variants** | | | | | | | |
| Hb Lepore | | 10 (0.12) | 84.9 ± 5.5 | 2.5 ± 0.8 | 4.6 ± 5.5 | Hb Lepore 8.7 ± 2.1 | - |
| Hb J variant | | 7 (<0.1) | 70.3 ± 8.1 | 2.3 ± 0.7 | 0.8 ± 0.4 | Hb J 27.1 ± 7.9 | - |
| Hb H (–/-α) | | 3 (<0.1) | 94.2 ± 0.6 | 1.3 ± 0.5 | 0.4 | Hb H 4.1 ± 1.3 | - |
| Hb N-Seattle | | 2 (<0.1) | 69.3 | 1.7 | 0.2 | Hb N 28.8 | - |
| Hb Hope | | 1 (<0.1) | 37.6 | 4.0 | - | Hb Hope 58.4 | - |
| Hb C | | 1 (<0.1) | 53.7 | 5.3 | 0.9 | Hb C 40.1 | - |
| Total | | 1971 (23.18) | | | | | |

* HPFH = Hereditary Persistence of Fetal Hemoglobin.

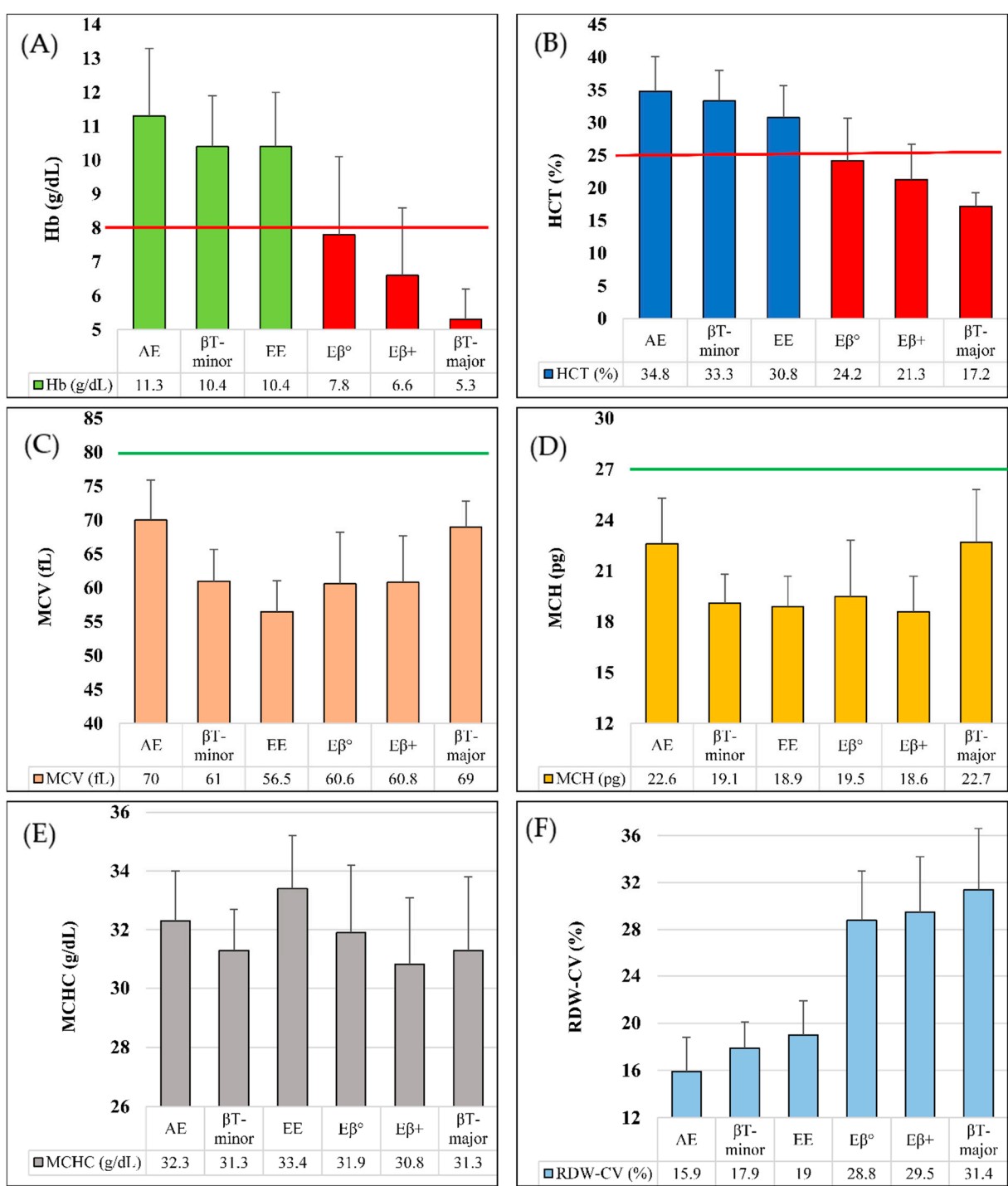

**Figure 1.** Graphical comparison in different types of thalassemia and hemoglobinopathy. (**A**) Mean Hb concentration (g/dL), severe anemic cutoff point shown in red line; (**B**) Mean HCT (%), severe anemic cutoff point shown in red line; (**C**) Mean MCV (fL), normal cutoff point shown in green line; (**D**) Mean MCH (pg), normal cutoff point shown in green line; (**E**) Mean MCHC (g/dL); (**F**) Mean RDW-CV (%). AE = Hb E (Heterozygous); βT-minor = β-thalassemia (Heterozygote); βT-major = β-thalassemia (Homozygote); EE = Hb E (Homozygous); Eβ° = Hb E/β⁰-thalassemia; Eβ⁺ = Hb E/β⁺-thalassemia.

**Table 2.** Erythrocytic variables for common inherited Hb disorders.

| Inherited Hb Disorders | Age * (Years), Median (IQR) | RBC ($10^6$/μL) Mean ± SD | Hb (g/dL) Mean ± SD | HCT (%) Mean ± SD | MCV (fL) Mean ± SD | MCH (pg) Mean ± SD | MCHC (g/dL) Mean ± SD | RDWCV (%) Mean ± SD |
|---|---|---|---|---|---|---|---|---|
| Hb E (Heterozygote) (*n* = 907) | 24 (5.5–36) | 5.0 ± 0.6 | 11.3 ± 2.0 | 34.8 ± 5.3 | 70.0 ± 5.9 | 22.6 ± 2.7 | 32.3 ± 1.7 | 15.9 ± 2.9 |
| β-thalassemia (Heterozygote) (*n* = 714) | 27.5 (9–40) | 5.5 ± 0.8 | 10.4 ± 1.5 | 33.3 ± 4.7 | 61.0 ± 4.7 | 19.1 ± 1.7 | 31.3 ± 1.4 | 17.9 ± 2.2 |
| Hb E (Homozygote) (*n* = 135) | 26 (12–40) | 5.5 ± 0.9 | 10.4 ± 1.6 | 30.8 ± 4.9 | 56.5 ± 4.6 | 18.9 ± 1.8 | 33.4 ± 1.8 | 19.0 ± 2.9 |
| Hb E/β⁺-thalassemia (*n* = 101) | 7 (2.5–22) | 3.6 ± 1.0 | 6.6 ± 2.0 | 21.3 ± 5.4 | 60.8 ± 6.9 | 18.6 ± 2.1 | 30.8 ± 2.3 | 29.5 ± 4.7 |
| Hb E/β⁰-thalassemia (*n* = 33) | 5 (1.25–17) | 3.9 ± 0.9 | 7.8 ± 2.3 | 24.2 ± 6.5 | 60.6 ± 7.6 | 19.5 ± 3.3 | 31.9 ± 2.3 | 28.8 ± 4.2 |
| β-thalassemia (Homozygote) (*n* = 9) | 0.7 (0.5–1) | 2.4 ± 0.6 | 5.3 ± 0.9 | 17.2 ± 2.1 | 69.0 ± 3.8 | 22.7 ± 3.1 | 32.0 ± 2.5 | 31.4 ± 5.2 |
| Hb D-Punjab (Heterozygote) (*n* = 16) | 6.5 (2–24.5) | 4.7 ± 0.8 | 10.2 ± 2.3 | 31.1 ± 5.2 | 66.8 ± 9.1 | 21.8 ± 4.2 | 32.5 ± 2.5 | 18.9 ± 4.2 |
| HPFH (Heterozygote) (*n* = 13) | 28 (22–41) | 4.9 ± 1.1 | 11.0 ± 2.4 | 35.9 ± 7.2 | 74.2 ± 10.1 | 22.8 ± 2.9 | 30.8 ± 1.1 | 19.5 ± 3.4 |
| Hb S (Heterozygote) (*n* = 9) | 24 (4–28) | 5.1 ± 0.4 | 12.2 ± 2.8 | 36.9 ± 7.3 | 73.0 ± 12.6 | 24.1 ± 4.7 | 32.9 ± 1.9 | 16.1 ± 4.1 |
| Hb Lepore (Heterozygote) (*n* = 10) | 24 (5.5–26) | 5.4 ± 1.0 | 11.0 ± 2.7 | 35.1 ± 7.8 | 65.4 ± 5.2 | 19.9 ± 1.8 | 30.8 ± 1.5 | 20.2 ± 3.7 |
| Hb E/Hb S (*n* = 5) | 65 (6–66) | 3.1 ± 1.2 | 9.5 ± 1.5 | 28.0 ± 4.6 | 83.0 ± 22.3 | 28.1 ± 9.1 | 34.5 ± 0.9 | 20.4 ± 7.7 |
| Hb H (–/-α) (*n* = 3) | 20 (1–45) | 4.5 ± 0.3 | 8.6 ± 0.2 | 29.9 ± 1.0 | 66.8 ± 1.9 | 19.2 ± 1.5 | 28.7 ± 1.5 | 22.6 ± 4.9 |

* Age at the time of diagnosis.

Single case of Hb C (heterozygote), Hb Hope, Hb S/β⁺-thalassemia, Hb D-Punjab (homozygote), Hb D-Punjab/β⁺-thalassemia, Hb S/Hb D-Punjab and HPFH (homozygote) each was observed as well. Hb C (heterozygote) case has 40.1% Hb C at Z2 position in capillary zone electrophoresis (CZE) with 53.7% Hb A, 5.3% Hb A2 and tracer amount (0.9%) of Hb F. The red cell indices like MCV, MCH, and MCHC are 59.4 fL, 20 pg, 33.6 g/dL, respectively along with RBC (5.6 × $10^6$/μL), Hb (11.1 g/dL), HCT (33%). There was one sickle cell trait with β⁺-thalassemia so far where 66.1% Hb S, 7% Hb A, 4.8% Hb A2 and rest of the percentage for Hb F were observed. There was a case of double heterozygote condition of Hb S (43.3%) with Hb D-Punjab (50.8%) (Figure 2A). The possible case of Hb Hope variant (58.4%) has Hb A and Hb A2 with 37.6% and 4.0%, respectively (Figure 2C). Its Hb, MCV and MCH are 9.5 g/dL, 71.3 fL and 22.4 pg, respectively. The only incident of homozygotic HPFH had 100% Hb F with Hb 11.6 g/dL, MCV 60.1 fL, MCH 18.8 pg and RDW-CV 22.3% (Figure 2B).

One of two Hb N-Seattle cases showed red cell indices (MCV = 87.3 fL, MCH = 29.2 pg, RDW-CV = 15.3%) with Hb = 13.1 g/dL, while the other featured differently (MCV = 64.2 fL, MCH = 18.1 pg, RDW-CV = 18.3%) with Hb = 9.4 g/dL. Their mean Hb N was 28.8%. The zone at Z14 clearly designates Hb N-Seattle variant (Figure 2D) (See Supplementary file Table S1).

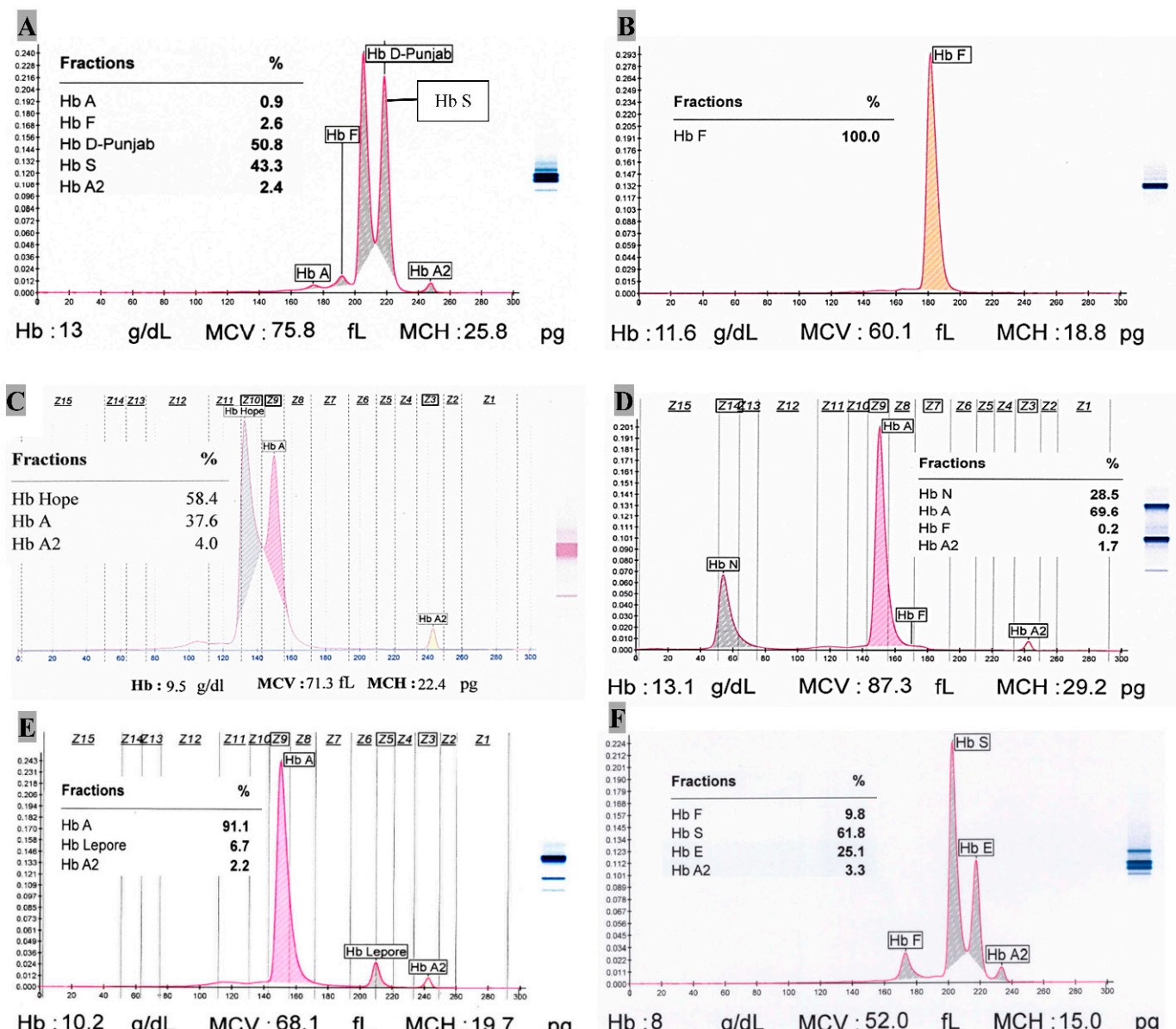

**Figure 2.** Capillary zone electrophoresis of some Hb variants. (**A**) Hb S/HB D-Punjab (Double heterozygote); (**B**) Hereditary Persistence of Fetal Hemoglobin (Homozygote); (**C**) Hb Hope; (**D**) Hb N-Seattle (Heterozygote); (**E**) Hb Lepore (Heterozygote); (**F**) Hb E/Hb S (Double heterozygote).

## 4. Discussion

We have identified the Hb N-Seattle variant (*n* = 2) which has not been previously reported in Bangladesh, as far as we know. However, Hb N-Seattle variant has been identified and published in India as a case report in 2017 [32]. In our study, one case showed normal red cell indices with normal hemoglobin concentration, while the other one presented microcytic hypochromic behavior with moderately anemic condition (Hb = 9.4 g/dL). Their mean percentage of Hb A2 (1.65%) was significantly lower than the normal levels (2–3%) [33].

We have found Hb D-Punjab variant in different patterns like heterozygotic (*n* = 16), homozygotic (*n* = 1) forms and compound heterozygotic condition with β-thalassemia (*n* = 1), Hb S (*n* = 1). Hb D-Punjab in heterozygous condition with Hb A generally presents no significant clinical features [34]. However, we have found the mean MCV as 66.8 fL ± 9.1 (95% CI, 62.0–71.7). Lower MCV reflected their microcytic feature, but not such a severe anemic condition that would require blood transfusion as the mean Hb was 10.2 g/dL ± 2.3

(95% CI, 9.0–11.4). We have identified a homozygotic Hb D-Punjab case with 94.9% Hb D-Punjab, 3.4% Hb A2 and 1.7% Hb F. Generally, Hb A cannot be found in the homozygotic state of Hb D-Punjab [35]. The red cell indices (MCV = 77.2 fL, MCH = 26.8 pg, MCHC = 34.7 g/dL, RDW-CV = 14.8%), Hb concentration (12.6 g/dL) showed no symptomatic features [34]. On the other hand, there was a case of co-inheritance pattern of Hb D-Punjab with $\beta^+$-thalassemia trait which represented microcytic (MCV = 56 fL), hypochromic (MCH = 18.3 pg) and moderate anemic (Hb = 9.9 g/dL) behaviors. Moreover, it contained higher Hb A2 (5.2%), 4.1% Hb A (that excluded $\beta^0$-thalassemia [36]), 89.3% Hb D-Punjab and 1.4% Hb F. However, the percentage of Hb A2 as the only reliable diagnostic parameter while differentiating Hb D-Punjab/$\beta$-thalassemia and Hb D-Punjab (homozygote) is not confirmed unanimously in all relevant studies [37–39].

We have found the double heterozygosity of Hb E/Hb S ($n = 5$), Hb S/Hb D-Punjab ($n = 1$) and Hb S/$\beta^+$-thalassemia ($n = 1$). In Hb S/Hb D-Punjab, Hb D (50.8%) forms a little bit higher proportion of total hemoglobin than does Hb S (43.3%) concordant with other studies [40]. Hb S with $\beta^+$-thalassemia had 7% Hb A and 22.1% Hb F which excluded the chance of being Hb S/HPFH as the latter doesn't have any Hb A. Moreover, Hb S comprises more than 50% (here, 66.1%) of total hemoglobin, in contrast to a heterozygotic sickle cell having <50% Hb S. The Hb S/$\beta^+$-thalassemia case had microcytic (MCV, 64.7 fL), hypochromic (MCH, 22.1 pg) features along with slightly raised RDW-CV (17.6%), similar to other studies [40,41]. Unlike the previous one, Hb A is totally absent in Hb E/Hb S. Despite having lower frequency ($n = 5$), Hb E/Hb S cases were reported for having variable phenotypes where some showed asymptomatic features and others developed sickling-related complications, similar to the findings of related studies [42,43].

There are some possible reasons behind the very low number of $\beta$-thalassemia patients in our study. As we excluded all cases with a recent history of blood transfusion (three months prior to sample collection), most of the $\beta$-thalassemia major cases could not be included. $\beta$-thalassemia patients show their symptoms of thalassemic criteria when they are 3 to 6 months of age [44]. We have found total of 9 samples with median age of less than 9 months. Maximum cases of such patients who have already been transfused by blood showed otherwise information in capillary zone electrophoresis of Hb due to the presence of high Hb A coming from donors' blood. $\beta$-thalassemia major displayed a higher percentage of Hb F resulting in severe anemic condition with very low Hb concentration (5.3 g/dL $\pm$ 0.9 g/dL; 95% CI, 4.6–6.0).

In the diagnosis of the heterozygote condition of $\beta$-thalassemia, the proportion of Hb A2 relative to the other hemoglobins is clinically important. In certain cases, Hb A2 variants may also be present. In such cases, the total Hb A2 (Hb A2 and Hb A2 variant) needs to be considered for the diagnosis of $\beta$-thalassemia [45,46]. The average value of HbA2 was 5.2% in $\beta$-thalassemia (heterozygote) compared to the cutoff value of >3.5% Hb A2 (at least 4% [39]) as a diagnostic hallmark for $\beta$-thalassemia (heterozygote) mentioned in other studies [8,47,48].

Both heterozygote and homozygote conditions of Hb E along with $\beta$-thalassemia trait and Hb E/$\beta$-thalassemia consist of nearly 96% among total 1973 samples having inherited Hb disorders. All of them were microcytic in nature as their mean MCVs were less than 80 fL. However, homozygous Hb E had the lowest mean MCV (56.5 fL $\pm$ 4.6, 95% CI, 55.7–57.3). This condition of having the lowest MCV might be co-inherited with different forms of alpha thalassemia [49]. Only homozygotic features of $\beta$-thalassemia and Hb E/$\beta$-thalassemia cases showed the feature of severe anemic condition. $\beta$-thalassemia (homozygote) and Hb E/$\beta$-thalassemia ($\beta^+$- and $\beta^0$- type) samples were more anisocytic than Hb E (Heterozygote and homozygote) and $\beta$-thalassemia trait in terms of corresponding mean RDW-CV. In the electropherogram, Hb E homozygosity is distinguishable from Hb E/$\beta^\circ$-thalassemia in terms of the percentages of Hb F and Hb E. The latter contains 40–60% Hb F with the remainder being Hb E and Hb A2, whereas 85–95% Hb E is predominantly seen in Hb E homozygosity [43]. Similar findings were observed in our

study as well. However, there were no significant differences found in mean MCVs, MCHs, MCHCs and RDW-CVs between Hb E/$\beta^\circ$-thalassemia and Hb E/$\beta^+$-thalassemia.

Hemoglobin Lepore (named after an Italian family where it was first identified [50]) caused by unequal crossing-over of the β and δ genes [9] is one of the prominent types of $(\delta\beta)^+$-thalassemia [43]. Hb Lepore trait showed reduced Hb A2 (2.5% $\pm$ 0.8; 95% CI, 2.0–3.1) levels along with other hematological pictures characterized by microcytosis (mean MCV, 65.4 $\pm$ 5.2; 95% CI 61.7–69.1) and hypochromia (Mean MCH, 19.9 $\pm$ 1.8; 95% CI, 18.7–21.2).

This study was neither any epidemiological nationwide survey nor was the study population chosen randomly. We have some limitations in this study like, not all samples could be examined for co-existing alpha-thalassemia. The two common α+-thalassemia deletion genes (e.g., -α3.7 and -α4.2) together with five α°-thalassemia deletion genes (e.g., –SEA, –THAI –FIL, –MED, and -α20.5) generally diagnosed by gap-PCR followed by hybridization technique [43] cannot be done in this study. In capillary zone electrophoresis, the hemoglobin variants (e.g., Hb H, Hb Constant Spring) representing α-thalassemia can be detected in a designated zone. However, Hb H and Hb Constant Spring variants are unstable and may degrade rapidly. At the molecular (DNA) level, Hb H most commonly results from co-inheritance of α°-thalassemia with α+-thalassemia deletions [43,51]. So, by capillary zone electrophoresis, we could not detect many α-thalassemia mutations. According to Riou et al. 2018 [52], the reproducibility of the relative migration position of Hb Hope at zone 10 showed excellent precision, similar to our study. However, definitive identification cannot be achieved from a single technique like CZE unless corresponding DNA analysis or amino acid sequencing can be done simultaneously. Besides, we have found 7 cases of Hb J variants at different zones (Z11 to Z13). In order to precisely diagnose those variants, direct DNA sequencing is highly recommended as well.

## 5. Conclusions

This facility-based study depicts the diversity and heterogeneity of thalassemia and hemoglobinopathy in Bangladesh. The article aims to gather and improve knowledge about such inherited hemoglobin disorders among hematologists, general practitioners, young internists, human geneticists, and laboratory-based physicians in their corresponding clinical practices and research. Moreover, this broad and detailed spectrum will ultimately be helpful in implementing nationwide thalassemia management and strategy policy in Bangladesh.

**Supplementary Materials:** The following supporting information can be downloaded at: https://www.mdpi.com/article/10.3390/thalassrep13020012/s1, Supplementary Table S1: Potential Hb variants in each zone in Sebia Capillary-2 electrophoresis.

**Author Contributions:** Conceptualization, A.H. and D.M.; methodology, A.H., J.A. and M.A.; software, A.H.; validation, A.H., D.M. and J.A.; formal analysis, A.H. and J.A.; investigation, A.H., J.A., M.A., B.C.C., A.R.S., P.K.D., K.A.B.H., M.Z.I. and R.A.; resources, J.A., M.A., B.C.C., A.R.S. and R.A.; data curation, A.H., D.M., S.Z.U. and J.A.; writing—original draft preparation, A.H.; writing—review and editing, D.M. and S.Z.U.; visualization, A.H.; supervision, D.M. and S.Z.U.; project administration, D.M. All authors have read and agreed to the published version of the manuscript.

**Funding:** This research received no external funding. However, we are grateful to the Government of Bangladesh, Canada, Sweden and the UK for providing core/unrestricted support.

**Institutional Review Board Statement:** The study was conducted in accordance with the Declaration of Helsinki, and approved by the Research Review Committee and Ethical Review Committee belong to Institutional Review Board (IRB) of International Centre for Diarrhoeal Disease Research, Bangladesh (icddr,b) (protocol code PR-22053 and date of approval: 12 June 2022).

**Informed Consent Statement:** It is not applicable because the study protocol followed the retrospective secondary data analysis method based on the archived data of already analyzed specimens. And this manuscript doesn't contain anyone's individual data which will reveal any identity from any point of view.

**Data Availability Statement:** The original contributions presented in the study are included in the article/Supplementary Material. Further inquiries can be directed to the corresponding authors.

**Acknowledgments:** We thank Hafizur Rahman for his unyielding cooperation in this study. We gratefully appreciate the support of all staff of the Clinical Hematology and Cancer Biology Laboratory, icddr,b and Research Administration, icddr,b. Our special gratitude goes to Ananda Chandra Roy, Md. Mazed Ali, Din Mohammad, Md. Kabirul Islam.

**Conflicts of Interest:** The authors declare no conflict of interest.

## Abbreviations

CBC: Complete blood count; CZE: Capillary zone electrophoresis; CI: Confidence interval; SD: Standard deviation; HCT: Hematocrit; MCV: Mean corpuscular volume; MCH: Mean corpuscular hemoglobin; MCHC: Mean corpuscular hemoglobin concentration; RDW: Red cell distribution width; CV: Coefficient of variation; High-performance liquid chromatography (HPLC); HPFH: Hereditary persistence of fetal hemoglobin.

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
