# Peer review of "Spectrum of Thalassemia and Hemoglobinopathy Using Capillary Zone Electrophoresis: A Facility-Based Single Centred Study at icddr,b in Bangladesh"

_thalassrep, doi:10.3390/thalassrep13020012_

Round 1

Reviewer 1 Report

Abstract: lack of specification of Data: clinical data: not obtainable by capillary electrophoresis nor complete blood count, which clinical data was obtained?;  haematologic data: obtainable by complete blood count and by capillary electrophoresis (Hb-Analysis); (diagnosis can be made by the combination of these data or verfication by second methods)

the degree of which the the taken study population of about 8500 samples reflects the distribution in the country Bangladesh is not well explainded. Age and sex ist not specified (if due to anonymization, than statement nessecary) , What was the reason of referrel to the centre?, is it the only centre in Bangladesh or are there more laboratories?, are patients included repeatedly,  (double examinations?), who was contacted for the information about a history of  transfusion, where samples than all excluded? (e.g. this gives bias  to the estimated prevalence of severe hemoglobinopathies, see discussion)

2.3.2 wrong title (blood count and CE)  sample preparation (mixing by 15-20rpm???) for analysis in Sysmex XT 1800 seem questionabel: why paraemter as hemoglobin-concentration, counts of leukocytes, erythrocytes, thrombocytes were not reported, but the indices? (clear statement in the method section and reporting results in the results section)

2.3.5 reference values of all parameters (in generally age and sex dependant) applied should be given, decision breakpoints and cut-off should be clearly stated. the cut-off for beta-Thalassemia heterozygous is method dependent and can have fals positiv and false netative (other causes of elevated hbA2, iron deficiency to lower HbA2 also in thalassemia trait). cut-off applied in the study should be stated (eg. 3.2% or 3.5% or higher) attention: a cut-off ist not the mean of studied samples, but can give bias to the statistics

iron deficiency ist bnor irrelevant bat has not nessecarely be shown and explaind in the present study. (wording!)

iron deficieny and co-existing alpha thalaseamia have both effects on Hb-variant quantities/percentages, this seems neglected in the paper

3. results: all Hb-percentages given with 3 digits after the decimal point, this is not appropriate to the precision of hb-analysis by CE, one digit after the decimal point is adaequate (and better readable).

How diagnis, or naming of variants was done if zone are not displayed ? This happens when no HbA is present in the sample (eg. HbD/HbS and HbF only (fig 2 A and B) was any of the diagnisi verified by second method (HPLC, molecular diagnosis?) migration in zones are only hints and diagnosis can only be presumptive.

taxonomy of alpha-thalassemia is not correct, the precence of HbH as an unstable variant can have different genetic background. deltabeta-thalassemia is not equivalent to HPFH and Hb Lepore

fig. 1 Hb-conc. MCV, MCH, MCHC, RDW-CV  given with 3 (the latter 2) digits after the decimal point, this is not appropriate to the precision, one is enough

HbD Punjab heterozygous with MCV lower than 70fl Co-existing iron deficieny and /or alpha-thalassemia?

table 2: HbH interpetation as heterozygous is not applicable (aa/aa, -a/-a/, --/aa, --/-a)

I reccomend major revision of the results,

Reviewer 2 Report

The manuscript needs to be edited for English style. You should state how you distinguished E Beta zero thalassaemia from E homozygosity. There are some minor errors that need correction: ul needs the correct Greek symbol for microlitre; PCV could be deleted  - just use haematocrit; an MCH less than 27 pg is useful for suggesting possible thalassaemia but it does not correlate with hypochromia - it is the MCHC that does that. In figure I it would be better in the graphcs to use more instantly intelligible abbreviations, e.g. AE for E heterozygosity, EE for E homozygosity and so on, and a beta symbol not a 'B'. How was the identity of N-Seattle, Hope etc. confirmed. Capillary electrophoresis alone would not normally be taken as confirmation. Some of the references are incomplete, e.g. reference 19. Some references are unnecessary, e.g. several text books. The paper could be shortened.

Round 2

Reviewer 1 Report

Thank you for revising the manuscript. Now it is much better in understanding and more precise discription of results. A very interesting and impotant work.

remark: fig 2 A, B and F show now zones, what gives hint to the lack of detecting HbA. The zones can anly be shown, when HbA is present in the sample. manufacturer gives the recommendation adding HbA (by diluting with a normal sample containing HbA) to the sample, so that zones can be applied and than presumptive diagnosis of the HbVariant based on the zones can be made. The percentages of the Hb variants is taken from the original non diluted sample pherogram.

cut of for diagnosis heterozygopus beta-thalassemia only which was used in the study  is not given. only mean of 5.2%  (did I overlooked it)?

Discussion need to be completed for all hemoglobinopathies found in the study - as in most cases presumtively diagnosed. The found mean MCV (56,5fl) for HbE homzygous is to low in comparison with most known publications. A good overview ist given by Fucharoen and Wetherall in 2012 "The hemoglobin E thalassemias", doi: 10.1101/cshperspect.a011734 . It is explainable by the limitytion that not all samples could be examined for co-existing alpha-thalassemia.

As HbH is a very unstable hemoglobin, which occurs in -a/-a, --/aa and sure in --/-a alpha thalassemia conditions and it may degrade rapidely (either disapear or show in a different zone) and therefore ist not in all cases the reliable marker for including or excluding alpha-thalasemia, especially in co-existing state. Hb Constant Spring as is unstable and has only a expected % of 1 to 10 % is often ovelooked in pherograms .... (Did you detected any HbCS?)

despite this limitation results remain very interesting, but should e.g. for HbE heterozygous and homozygous be clearly be described as including all possible combinations of alpha-thalasemias and therefore lowering mean percentages of HbE in heterygous HbE, and lowering MCV and MCH in those cases.

a limitation also is that the study population is not exactly representative to the whole Bangladeshi population.

kind regards
